# Simulated Clinical Stations in Quality and Patient Safety in a Primary Care Setting

**DOI:** 10.3390/healthcare13131501

**Published:** 2025-06-24

**Authors:** Yoseba Cánovas Zaldúa, Sonia Martín Martín, Jordi Serraboguña Bret, Eduard Hermosilla Perez, Ermengol Coma Redon, Sara Rodoreda Noguerola

**Affiliations:** 1Directorate of Primary and Community Care, Catalan Institute of Health (ICS), 08035 Barcelona, Spain; ycanovas.bcn.ics@gencat.cat (Y.C.Z.); jserrabogunab.germanstrias@gencat.cat (J.S.B.); srodoreda.mn.ics@gencat.cat (S.R.N.); 2Passeig de Sant Joan Primary Care Centre, Catalan Institute of Health (ICS), 08035 Barcelona, Spain; 3Faculty of Medicine, Universitat Autònoma de Barcelona (UAB), Cerdanyola del Vallès, 08193 Barcelona, Spain; 4Primary Care Services Information Systems (SISAP), 08007 Barcelona, Spain; ehermosilla@gencat.cat (E.H.P.); ecomaredon@gencat.cat (E.C.R.)

**Keywords:** primary care, patient-centered care, patient simulation, clinical practicum, simulation training

## Abstract

Background/Objectives: Clinical simulation-based training has become established as an effective strategy to improve healthcare quality and patient safety. This pre–post observational study presents an innovative experience implemented at the Catalan Health Institute (ICS) through the evaluation of a training intervention based on Simulated Clinical Stations in Quality and Patient Safety. The main objective is to improve the competencies of Primary Care Teams (PCT) professionals in managing critical and urgent situations and to assess the impact of the intervention on knowledge and satisfaction using an immersive methodology based in active practice. Methods: More than 8.916 professionals participated in 285 training sessions at the Balmes Primary Care Center (CAP Balmes) simulation center (Barcelona). Knowledge data were collected before and after the training, along with satisfaction levels, showing significant improvement. Results: The analyses show a significant improvement in the knowledge acquired and a high level of participant satisfaction, reinforcing the value of clinical simulation as a key training tool. Conclusions: The study reinforces clinical simulation as an essential, scalable, and adaptable educational tool across different healthcare settings, establishing itself as a key resource for the continuous training of healthcare professionals.

## 1. Introduction

Healthcare organizations around the world are increasingly aware of the importance of training their professionals in patient safety and quality of care, ensuring that their clinical practice is both optimal and safe [1,2,3].

Multiple studies have demonstrated that simulation training significantly improves professionals’ clinical skills and reduces errors [4,5,6,7]. In particular, it enhances both technical and non-technical skills, including communication, leadership, and the management of emotional pressure—key elements in primary care settings.

Patient safety and quality are considered essential pillars in healthcare delivery [8,9,10]. Clinical simulation has been firmly established as an effective methodology for improving these competencies, as indicated by several recent studies [2,3,11,12]. In this context, the need for improvement has led to the implementation of innovative strategies—such as clinical simulation—with the specific aim of assessing its impact in Primary Care Teams (PCT). Nevertheless, it is important to recognize that simulation, although it striving to replicate reality as closely as possible, does not fully reproduce actual clinical environments. Therefore, its applicability is maximized when training is conducted periodically and integrated into ongoing professional development.

This study, promoted by the Primary and Community Care Directorate of the Catalan Institute of Health (ICS), analyses the impact of a training intervention through simulated clinical stations, aiming to equip Primary Care Team (PCT) professionals to manage actual critical situations. It also seeks to foster an active and meaningful learning environment that replicates routine clinical practice.

Over the past eight years, all Primary Care Team professionals—including non-clinical staff—have participated in training initiatives as part of the quality accreditation process promoted by the Department of Health, with the aim of fostering a culture of safety and continuous improvement [13].

At the same time, this study hypothesizes that clinical simulation not only improves knowledge but also increases professional satisfaction, with the expectation of significant post-training gains.

## 2. Materials and Methods

### 2.1. Target Population

The study is based on a prospective observational pre–post design without a control group and follows the general guidelines of the STROBE statement for observational studies. A total of 8.916 professionals from 186 PCT of the Catalan Institute of Health (ICS) participated.

Each team consisted of an average of 48 professionals, organized into multidisciplinary groups that included physicians, nurses, and administrative staff. To ensure full staff participation, some teams required more than one training day, resulting in a total of 285 training sessions. This structure allowed for broad coverage of the various professional profiles within the teams. Details of the number of participants by category are shown in Figure 1, and specific examples by station and group are summarized in Table 1.

The target population includes all professionals of the PCT who progressively attend a 7 h training session at the CAP Balmes simulation center, located in the basement of the corporate headquarters in Barcelona. Participants are organized into different multidisciplinary groups, mainly composed of physicians, nurses and administrative staff, with the possible inclusion of other professional profiles within the team.

### 2.2. Methods and Project Planning

Each group rotates consecutively through nine simulated clinical scenarios, each following a standardized script [8,9,12,14,15,16,17], which are as follows:(1)Basic/Immediate Life Support (BLS/ILS) with Automatic External Defibrillator (AED) or Cardiopulmonary Resuscitation (CPR) Code.(2)Sepsis Code.(3)Acute Myocardial Infarction (AMI) Code.(4)Stroke Code.(5)Hand Hygiene.(6)Needlestick injury (biological accident).(7)Airway Obstruction by Foreign Body (OVACE).(8)Suicide Code.(9)Violence Against Healthcare Professionals.

The training combines theory and practice through clinical stations with simulated patients (professional actors specifically trained for each case). This methodology offers a safe learning environment, promoting teamwork, assertive communication, and leadership. Participants are not informed in advance of the simulation scenarios they will encounter; this information was disclosed only on the day of the session.

### 2.3. Evaluation and Analysis Process

At each station, there is an observer from the participating team itself, who times the session (15 min of simulation plus 5 min of debriefing), and conducts a structured observation using a checklist, providing immediate feedback to the group. Tailored checklists were developed for each clinical station, derived from institutional protocols and clinical consensus, assessing competencies such as recognition of critical signs, decision-making, teamwork, and effective communication. At the end of the rotation, a plenary session is held to share results and observations.

Knowledge is assessed using two tests (pre- and post-training) scored from 0 to 10, with the same set of questions. Satisfaction is assessed through a survey with a 0 to 6 scale. The knowledge test includes 16 closed-ended questions (multiple-choice and checkbox format), developed by a team of clinical training experts, with an estimated completion time of 10 min. Sample questions are shown in Table 2.

Satisfaction is assessed through a survey containing 76 items grouped into the following categories (organization, methodology, applicability, and station-specific satisfaction) utilizing a 6-point Likert scale.

All data from both the knowledge and satisfaction questionnaires are collected through forms accessible via QR code. Prior to the training session, participants are provided with detailed information about the activity by email, including a notice about the use of their data for research purposes.

For statistical analysis, means and standard deviations were calculated for continuous variables, and absolute frequencies and percentages for categorical variables. Student’s *t*-test for independent samples was applied, given the anonymous nature of the surveys. Student’s *t*-test for independent samples was applied, considering the anonymous nature of the surveys. *p*-values (<0.001) and effect sizes (overall mean increase d = 1.2, considered large) are reported for each key outcome, and the results were stratified by professional categories (physician, nurse, administrative staff). A *p*-value < 0.05 was considered statistically significant. The analysis was performed using R software, version 4.4.3 (R Foundation for Statistical Computing, Vienna, Austria).

The data were anonymized, with no collection of identifying information, and therefore no ethics committee approval was required. The study complies with the General Data Protection Regulation (GDPR 2016/679) and follows the principles of the Declaration of Helsinki.

## 3. Results

From October 2022 to the time of writing, 186 PCT have participated, totaling 285 training sessions. This figure reflects that teams with a larger number of professionals required more than one session to ensure full staff participation. In total, 8.916 professionals participated.

The distribution by professional categories, presented in Figure 1, shows that the largest group was nurses, accounting for 30.9% (2.753 participants), followed by physicians at 27.8% (2.481 participants) and administrative staff at 23.1% (2.063 participants).

In terms of knowledge, participants showed a significant improvement, with all results being statistically significant (Figure 2). Before the training, the overall mean score was 5.1/10. Following the training, the mean score increase by 2.5 points to 7.6/10, representing a 49% improvement (*p* < 0.001).

For individual stations (Figure 2), the following was found:BLS/AED-CPR Code: improved from 6.2/10 pre-training to approximately 8.0/10 post-training, representing a 29% increase.AMI and Stroke Code: improved from 5.9/10 pre-training to approximately 8.0/10 post-training, representing a 36% increase.Suicide Code: improved from 4.3/10 pre-training to approximately 7.5/10 post-training, representing a 74% increase.Sepsis Code: improved from 4.1/10 pre-training to approximately 7.5/10 post-training, representing a 83% increase.

Over the past eight years, the organization’s primary care teams have been progressively involved in structured quality accreditation processes led by the regional health department. As part of this initiative, all professionals—including administrative and support staff—have received basic training in key patient safety protocols. However, the intensity of this training has varied depending on the professional role. This historical background may partially explain the differences observed in baseline knowledge scores in this study. Non-clinical staff started from lower initial scores compared to clinical professionals, but their knowledge gains post-training were still statistically significant. These findings reinforce the value of inclusive, interdisciplinary simulation-based interventions and demonstrate that meaningful improvement is achievable across all professional profiles, regardless of their starting point.

By professional categories, physicians (Figure 3), had initial scores of approximately 7/10, which increased to scores equal to or above 8/10 after training. Among nurses (Figure 4), initial scores were around 6.5/10 in the AMI, Stroke and SVI Codes, exceeding 8/10 after training, while in suicide and sepsis codes, they started at 4.7/10 and 4.8/10, respectively, reaching 7.8/10 in both cases. For administrative staff (Figure 5), initial scores were around 5/10 in the AMI, Stroke and SVI Codes and around 3/10 in the suicide and sepsis Codes, later reaching 7/10 and 6.5/10, respectively.

Finally, in terms of overall satisfaction, 100% of the participants completed the survey, with scores ranging from 5 to 6 out of 6 across all evaluation items, reflecting a high level of satisfaction irrespective of professional category (Figure 6).

## 4. Discussion

The clinical simulation program at CAP Balmes represents an innovative and effective strategy for improving patient safety and quality of care in Primary Care setting. Training through simulated scenarios has reduced professionals’ stress and improved their coordination in critical situations; this is in line with Caballero [18], who emphasizes that simulation not only enhances clinical response in emergencies but also helps professionals manage stress, a crucial factor in high-workload primary care environments.

From an educational perspective, the work by Zunzarren [3] underscores the necessity of simulation as an active learning methodology in healthcare education. The author emphasizes the importance of reflective practice and debriefing to consolidate knowledge, which has also been integrated into the training at CAP Balmes, where participants receive structured feedback from the instructors.

The multidisciplinary training has contributed to strengthening teamwork and collaborative decision-making, which is consistent with the findings of Owen et al. [19] (pp. 51–57), who demonstrated that simulation improves communication across different levels of care and facilitates the implementation of advanced care plans. This characteristic is particularly relevant at CAP Balmes, where the involvement of diverse professional profiles has contributed to improved coordination in complex clinical situations.

In addition, the incorporation of a next-generation manikin has enabled a more accurate assessment of professionals’ performance, which is consistent with Sinclair et al. [20], who emphasize the importance of combining simulation with advanced technology to optimize learning outcomes. However, whereas Sinclair focuses on the advantages of e-learning, our results indicate that immersion in physical simulation environments exerts a greater impact on clinical responsiveness and professional confidence.

The use of professional actors has enhanced the realism of the simulation, although it represents a financial challenge. According to Eddy et al. [21], this training modality is essential for enhancing communication and collaboration in clinical practice. At CAP Balmes, the integration of actors has proven particularly valuable for managing scenarios involving patients at risk of suicide or incidents of violence directed at healthcare personnel, thereby promoting a more immersive and authentic learning experience.

The experience of CAP Balmes also supports the conclusions of Murphy et al. [6], who found that simulation in multidisciplinary teams enhances both the performance and efficiency of patient care. Murphy also highlights that simulation facilitates the identification of weaknesses in clinical care protocols. This has been evident in our program, where the sessions have enabled the identification of areas of improvement in internal communication and timely emergency response.

In this context, the study by Alonso-Peña and Álvarez [4] offers a systematic review of the effectiveness of clinical simulation as a training tool in healthcare education. The authors highlight that simulation improves not only technical skills but also non-technical competencies such as communication and leadership. This is fully consistent with our findings, as simulation at CAP Balmes has enhanced not only the technical competencies of healthcare professionals but also strengthened essential aspects such as team decision-making and conflict management under pressure. Furthermore, Alonso-Peña and Álvarez [4] emphasize the importance of post-simulation assessments to reinforce learning, which has been incorporated into our program through pre- and post-training tests.

Furthermore, it is important to highlight that all participating professionals, regardless of their clinical background, demonstrated substantial knowledge gains after the training. While non-clinical staff started from lower baseline scores, probably due to less extensive previous training, they also achieved notable improvement. This finding supports the feasibility of cross-professional simulation training. This outcome can be attributed to the fact that, for the past eight years, all primary care teams under the Catalan Health Department have been engaged in quality accreditation processes, which include compulsory training for all staff members, including administrative and support roles. This organizational context promotes a shared learning approach and further enhances the inclusiveness and effectiveness of the simulation program [13].

Despite these positive results, it is important to acknowledge certain limitations of our study. One of the main limitations is the absence of a control group, which prevents comparative analysis and limits the interval validity of results. Future studies should consider including a control group to strengthen the robustness and generalizability of the findings to other healthcare settings. Additionally, the voluntary nature of participant involvement and the lack of longitudinal follow-up are aspects that should be addressed in future evaluations to further enhance the validity of the results. Nevertheless, despite these limitations, it is important to emphasize that this study has conducted an in-depth analysis of the results obtained at CAP Balmes, focusing not only on comparison with the existing literature but also on identifying local specificities and their practical implications for the development of future simulation programs.

Finally, despite the cost associated with the simulation using professional actors, the highly positive results have justified the ICS investment, consolidating this methodology as a strategic training tool to improve patient safety and team cohesion in healthcare. Despite the particularities of the local context, the results appear generalizable to other similar primary care settings. In line with Alonso-Peña and Álvarez [4] (pp. 45–67), it is recommended to continue developing simulation strategies based on reflective practice and ongoing evaluation to ensure meaningful learning and better knowledge transfer to real clinical settings.

## 5. Conclusions

This study concludes that training programs on patient and quality of care using simulation, where professionals take on more active and participatory role, are a powerful tool for knowledge acquisition.

It is important to emphasize the need to train all Primary Care Team (PCT) professionals for clinical situations that may arise in daily practice, so that each can assume their leadership role when required.

Additionally, feedback from the simulated clinical stations can generate opportunities for improvement that should be leveraged by management teams through the development of targeted leadership strategies.

Furthermore, training with structured simulated clinical stations involving trained actors has received excellent evaluations from professionals.

Finally, this approach is considered an innovative and reproducible intervention for any team interested in adopting it.

## Figures and Tables

**Figure 1 healthcare-13-01501-f001:**
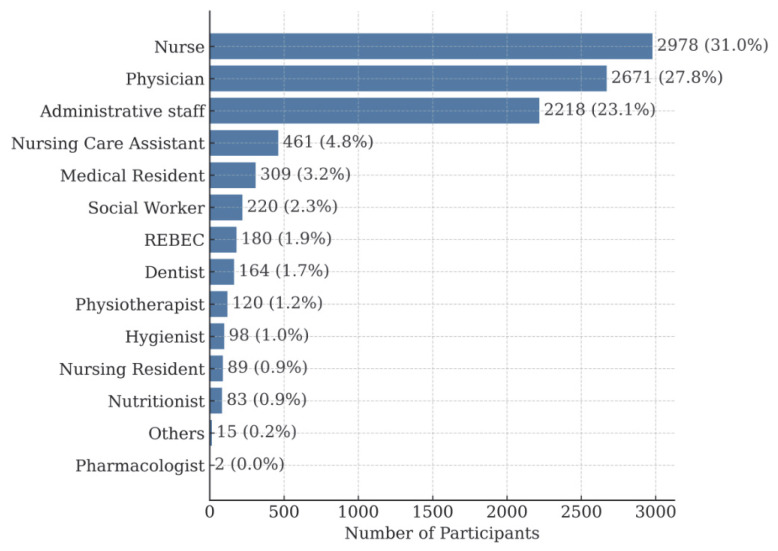
Distribution of participants by professional category.

**Figure 2 healthcare-13-01501-f002:**
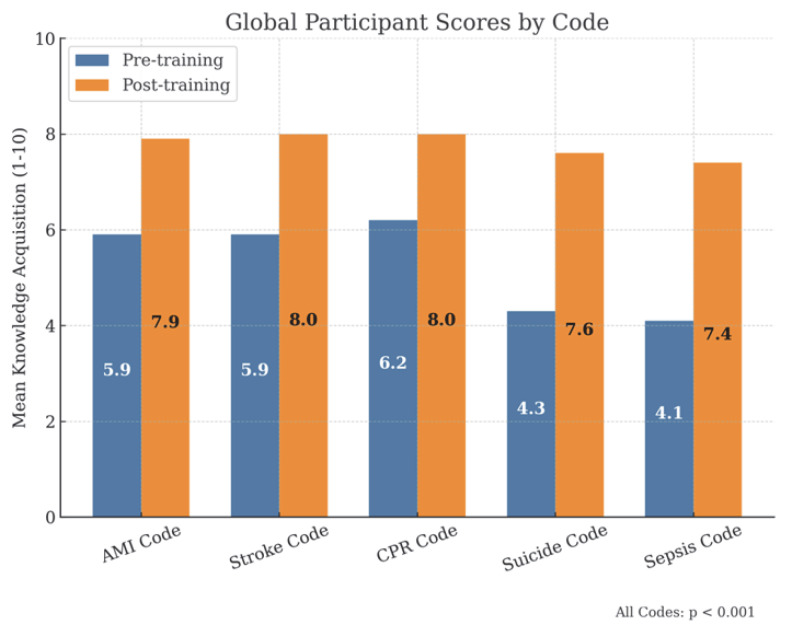
Global participant scores.

**Figure 3 healthcare-13-01501-f003:**
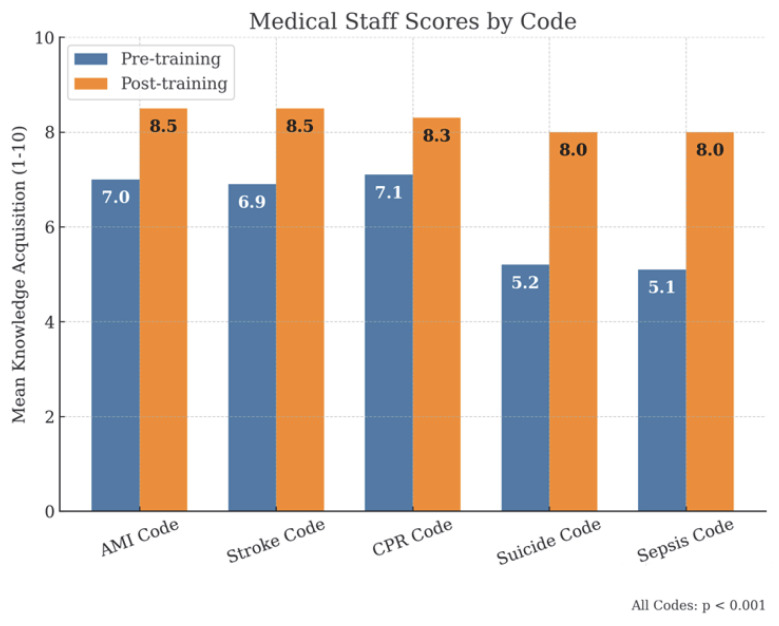
Medical staff scores.

**Figure 4 healthcare-13-01501-f004:**
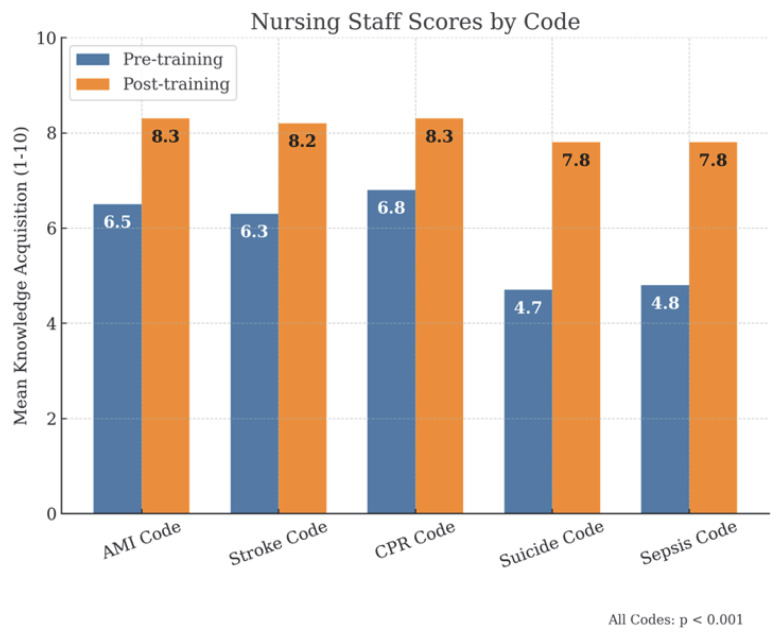
Nursing staff scores.

**Figure 5 healthcare-13-01501-f005:**
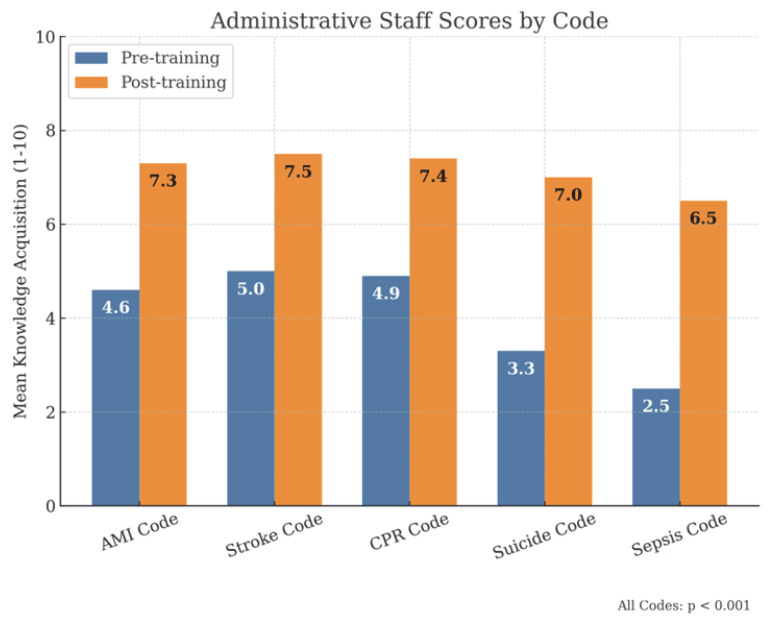
Administrative staff scores.

**Figure 6 healthcare-13-01501-f006:**
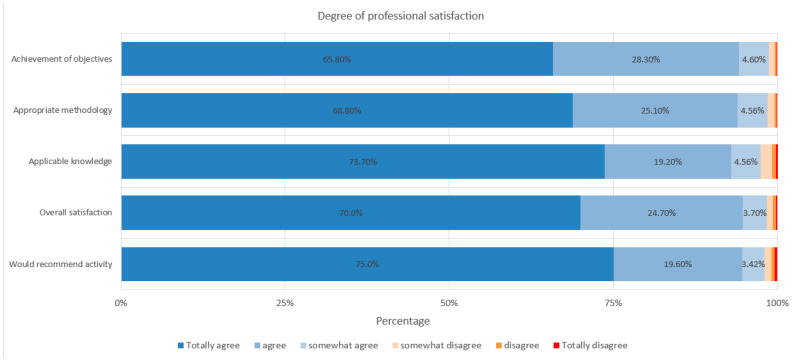
Degree of professional satisfaction.

**Table 1 healthcare-13-01501-t001:** Summary of participants by professional category, initial and final scores, and observed improvement.

Professional Category	Number of Participants (%)	Initial Mean Score (pre)	Final Mean Score (Post)	Increase (%)
Physicians	2.481 (27.83%)	7.0/10	≥8.0/10	+14%
Nurses	2.753 (30.88%)	6.5/10	>8.0/10	+23%
Administrative staff	2.063 (23.14%)	5.0/10 (AMI, Stroke, BLS/ILS), 3.0/10 (suicide, sepsis)	7.0/10, 6.5/10	+40%, +117%
Overall	8.916 (100%)	5.1/10	7.6/10	+49%

**Table 2 healthcare-13-01501-t002:** Examples of questions from the knowledge questionnaire.

Number	Example Question
1	Indicate your level of knowledge of the Sepsis Code (1 to 10).
2	Which products should you apply in case of accidental needlestick? (you can select more than one).
3	How many maximum hours determine the activation of the stroke code?
4	Do you know the MINI (International Neuropsychiatric Interview) test?
5	In the case of AMI, should the AMI code only be activated when there is ST-segment elevation? (true/false).

## Data Availability

The data supporting the findings of this study are not publicly available due to institutional privacy restrictions. All data analyzed were aggregated and anonymized.

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
