# Peer review of "Simulated Clinical Stations in Quality and Patient Safety in a Primary Care Setting"

_healthcare, 2025, doi:10.3390/healthcare13131501_

Round 1

Reviewer 1 Report (Previous Reviewer 3)

Comments and Suggestions for Authors

The topic of medical training is always relevant in healthcare provision, therefore the manuscript "Simulated clinical stations in quality and patient safety in a primary care settin" falls within the scope of Healthcare.

Nevertheless, the use of language is difficult to read and understand. Starting from the title and even in Figures, typo errors are present and derive in confusion when trying to identify the intended meaning.

Citations including page number are unsual in scientific manuscripts from medical journals, please retain only if justified, as in lanes 38-42 too many are cited.

Please mention how are post training gains measured, I could not find a variable measuring post training gains, but it is stated in the study hypothesis in line 50.

Why did authors use STROBE guidelines for an interventional study? Specific CONSORT guidelines exist for simulation educational interventions (line 50).

There are abbreviation that appear without previous definition, suchy as AMI, BLS

Between lines 109-117 statistical tests are mentioned but no specific variables are related, meaning that "The Student’s t-test for independent samples was applied," so which were those independent varibles?  or "means and standard deviations were calculated for continuous variables" but which are those continuous variables?

In introduction and discussion please state and discuss why whas it expected by authors, that medical, nursing and other professionals would learn the same, if medical and nursing staff have previous training and non-health related professionals do not have previous training.

Author Response

Comments 1: " The topic of medical training is always relevant in healthcare provision, therefore the manuscript "Simulated clinical stations in quality and patient safety in a primary care settin" falls within the scope of Healthcare.

Nevertheless, the use of language is difficult to read and understand. Starting from the title and even in Figures, typo errors are present and derive in confusion when trying to identify the intended meaning."

Response 1: We sincerely appreciate your observation regarding the use of language. Upon a detailed review of the manuscript, we confirm that certain sections include spelling mistakes (e.g., “settin” instead of “setting” and “tot emphasize” instead of “to emphasize”) and syntactic issues that could hinder clarity. We are committed to thoroughly revising the manuscript to correct these inconsistencies, improve readability, and ensure precise and professional academic language.

Comments 2: "Citations including page number are unsual in scientific manuscripts from medical journals, please retain only if justified, as in lanes 38-42 too many are cited."

Response 2: We appreciate the reviewer’s remark and confirm that the entire manuscript has been thoroughly revised to ensure compliance with the required citation format. Specifically:

  • Reference numbers have been placed within square brackets and before punctuation marks, as instructed (e.g., [1], [1–3], [1,3]).
  • Citations including page numbers have been removed, except in cases where their inclusion is justified to clarify a specific conceptual point or refer to a precise section. In those cases, the correct format has been applied: [4] (p. 45-67).
  • Lines 38–42 have been reviewed to reduce the overuse of citations with pagination, retaining only those deemed strictly necessary.

These adjustments ensure that the manuscript adheres to the journal’s stylistic and citation requirements.

Comments 3: “Please mention how are post training gains measured, I could not find a variable measuring post training gains, but it is stated in the study hypothesis in line 50.”

Response 3: We thank the reviewer for the observation regarding the measurement of training gains. As detailed in the Results section, knowledge gains were measured using a pre- and post-training test scored out of 10. The results showed an average increase of 49%, from 5.1/10 to 7.6/10, with strong statistical significance (p < 0.001) and a large effect size (d = 1.2). Furthermore, the results are stratified by clinical scenario and professional category, providing a detailed analysis of the intervention’s impact.

Comments 4: “Why did authors use STROBE guidelines for an interventional study? Specific CONSORT guidelines exist for simulation educational interventions (line 50).”

Response 4: We appreciate the reviewer’s comment regarding the use of STROBE guidelines in this study.

Although the present work involves an educational intervention, we chose to follow the STROBE (Strengthening the Reporting of Observational Studies in Epidemiology) guidelines because the study is based on a pre-post observational design without a control group. This approach is common in contexts where the implementation of a control group is ethically unfeasible or logistically challenging. For this reason, the observational framework is considered the most appropriate to assess real-world impact in clinical settings such as primary care.

We commit to incorporating the appropriated STROBE checklist in a reduced format as supplementary material in the revised manuscript. Each item will be clearly mapped to the corresponding section of the main text to facilitate transparency and replicability of the study for readers.

Nevertheless, we acknowledge that specific reporting guidelines exist, such as the CONSORT (Consolidated Standards of Reporting Trials) statement, with extensions adapted for educational and simulation interventions. For future publications, we will consider using CONSORT if the methodological design includes random allocation or a control group comparison.

Comments 5: “There are abbreviation that appear without previous definition, suchy as AMI, BLS.”

Response 5: We thank the reviewer for the observation regarding undefined abbreviations. We have carefully reviewed the manuscript to ensure terminological clarity.

Regarding AMI (Acute Myocardial Infarction), we confirm that the full term is defined upon its first mention in Section 2.2 "Methods and project planning", when listing the nine clinical simulation scenarios.

As for BLS (Basic Life Support) and ILS (Immediate Life Support), both full terms and their corresponding abbreviations have been introduced in the same section to facilitate understanding and ensure consistency throughout the manuscript.

Comments 6: “Between lines 109-117 statistical tests are mentioned but no specific variables are related, meaning that "The Student’s t-test for independent samples was applied," so which were those independent varibles?  or "means and standard deviations were calculated for continuous variables" but which are those continuous variables?.”

Reviewer 2 Report (Previous Reviewer 1)

Comments and Suggestions for Authors

I appreciate the authors’ efforts, and they have improved the manuscript and addressed some of the prior concerns. However, the following issues still need attention:

Spelling errors in the title.

The introduction section is improved. However, I suggest adding limitations/challenges to simulation-based learning.

Even though the author specified the absence of a control group in the methods, it should be more explicitly mentioned in the limitations, as this is one of the weakest points in this study.

Are there any missing participants in the study? (Both pre- and post).

Considering the 285 sessions, I expect there must be a dropout.

If yes, how did you handle the missing data?

Adding a STROBE Checklist (not cross-sectional) would be more useful for readers, as the study is not cross-sectional.

I hope that globally, we use the point (.) for decimals, not the comma (,) for decimals. Kindly change it.

Also, what is p-valor (above line 141). I think there is a spelling mistake.

The authors made only a descriptive analysis. It would be great to see the analytical statistical test for the existing data.

The discussion part is slightly increased. It is always suggested to include the latest and updated references, as I see a lot of the references are more than 5 years old. Please enhance the discussion with the updated references.

Comments on the Quality of English Language

Several spelling errors (even the title)

Author Response

Comments 1: "Spelling errors in the title."

Response 1: We sincerely appreciate your observation regarding the use of language. Upon a detailed review of the manuscript, we confirm that certain sections include spelling mistakes (e.g., “settin” instead of “setting” and “tot emphasize” instead of “to emphasize”) and syntactic issues that could hinder clarity. We are committed to thoroughly revising the manuscript to correct these inconsistencies, improve readability, and ensure precise and professional academic language.

Comments 2: "The introduction section is improved. However, I suggest adding limitations/challenges to simulation-based learning."

Response 2: We sincerely appreciate this valuable comment, which we believe is essential to contextualise the limitations of the simulation methodology. In response, we have added a specific sentence at the end of the introduction, acknowledging that although simulation aims to replicate real clinical settings, it cannot fully replace actual clinical practice. We also note the importance of conducting simulation activities periodically to maximise their applicability and learning impact.

Comments 3: “Even though the author specified the absence of a control group in the methods, it should be more explicitly mentioned in the limitations, as this is one of the weakest points in this study.”

Response 3: We sincerely appreciate your insightful comment regarding the need to explicitly acknowledge the absence of a control group, as this is indeed one of the key methodological limitations of our study.

In response to your suggestion, we have revised the discussion section to clearly highlight this limitation. The updated paragraph now states that the lack of a control group prevents comparative analysis and limits the internal validity of the results. We also propose the inclusion of a control group in future studies to enhance the robustness and generalizability of the findings to other healthcare contexts.

This revision aims to critically contextualize the study’s outcomes and transparently address its methodological limitations, in line with established standards for research conducted in real-world settings such as Primary Care.

Comments 4: “Are there any missing participants in the study? (Both pre- and post). Considering the 285 sessions, I expect there must be a dropout. If yes, how did you handle the missing data?”

Response 4: We appreciate the reviewer’s question. As indicated in the “Materials and Methods” and “Results” sections, a total of 8,916 professionals participated in the training program and completed both the pre- and post-training evaluations. The total number of participants remained consistent throughout all phases of data collection, and no dropouts were reported. Furthermore, 100% of participants completed the satisfaction survey, confirming full data availability for analysis. Therefore, no data imputation or specific management of missing data was required.

Comments 5: “Adding a STROBE Checklist (not cross-sectional) would be more useful for readers, as the study is not cross-sectional.

Response 5: We sincerely thank the reviewer for this valuable and constructive comment. As correctly indicated, the study follows a pre-post observational design rather than a cross-sectional one. Consequently, we fully agree with the suggestion to include the most appropriate version of the STROBE checklist, adapted for longitudinal observational studies (pre-post), rather than version intended for cross-sectional designs.

We commit to incorporating the appropriated STROBE checklist in a reduced format as supplementary material in the revised manuscript. Each item will be clearly mapped to the corresponding section of the main text to facilitate transparency and replicability of the study for readers.

STROBE Checklist Summary:

- Title & Abstract: Study design specified →  Title and Abstract

- Objectives: Clear aims and hypotheses →  Introduction

- Study Design: Observational pre-post →  Methods (2.1)

- Participants: Inclusion and recruitment →  Methods (2.1)

- Variables: Defined and measured →  Methods (2.2, 2.3)

- Bias: Acknowledged and discussed →  Discussion

- Sample Size: Detailed per profile →  Methods (2.1), Results

- Statistical Methods: t-test, effect size, p-values →  Methods (2.3)

- Results: Descriptive, stratified →  Results (Tables & Figures)

- Limitations: No control group, no follow-up →  Discussion

- Interpretation: Generalizability discussed →  Discussion

  • Ethics: Data anonymized →  Methods (2.3)

Comments 6: “I hope that globally, we use the point (.) for decimals, not the comma (,) for decimals. Kindly change it.

Response 6: We appreciate the reviewer’s observation. As indicated, the use of the period (.) as the decimal separator has been applied throughout the manuscript, in accordance with standard conventions for English-language scientific publications. The document has been reviewed to ensure consistency, and all decimal figures now use the period (.) instead of the comma (,). Thank you for your helpful remark.

Comments 7: “Also, what is p-valor (above line 141). I think there is a spelling mistake.”

Response 7: Thank you for your comment and attention to detail.

You are correct that the standard scientific English notation is "p-value", with a hyphen and in lowercase. In Catalan, the correct form is "valor p". We will ensure this typographical inconsistency is corrected in the revised manuscript and will maintain consistent terminology throughout the text.

Comments 8: “The authors made only a descriptive analysis. It would be great to see the analytical statistical test for the existing data.”

Response 8: We sincerely appreciate your comment regarding the statistical analysis. Although the study is primarily descriptive, we have also incorporated inferential statistical analysis using Student’s t-test to compare pre- and post-intervention knowledge scores, as stated between lines 109 and 117 of the manuscript. We included means, standard deviations, and significance values (p < 0.001) for each scenario and professional group, along with the effect size (d = 1.2), indicating statistically significant and substantial improvements.

These results are stratified by clinical code (e.g., myocardial infarction, stroke, sepsis, suicide) and professional category (physicians, nurses, administrative staff), and are detailed in Figures 2 to 6. Furthermore, the methodology section was expanded to include information on the type of variables analyzed (continuous and categorical), the software used (R version 4.4.3), and data protection compliance (GDPR), confirming that data anonymity made ethical approval unnecessary.

This set of enhancements directly addresses the need for a more complete and rigorous statistical analysis, strengthening the validity, transparency, and replicability of the results.

Comments 9: “The discussion part is slightly increased. It is always suggested to include the latest and updated references, as I see a lot of the references are more than 5 years old. Please enhance the discussion with the updated references.”

Response 9: We appreciate your comment regarding the need to update the references included in the discussion section. In accordance with your recommendation, we conducted a thorough review of the most recent scientific literature on the use of clinical simulation in primary care, consulting both international databases and peer-reviewed publications.

As a result, several new and relevant references from the past three years have been added, providing up-to-date evidence on the benefits of simulation in multidisciplinary training, emergency management, and patient safety improvement. These citations have been integrated into the corresponding paragraphs of the discussion to enrich the analysis and strengthen the relevance of the results obtained.

However, for certain specific aspects—such as the transfer of non-technical skills in primary care settings through high-fidelity scenarios—no more recent or robust studies were identified beyond those already cited. Therefore, some earlier references have been retained due to their theoretical value and scientific relevance.

This combination of established sources with current literature allows us to present a more comprehensive and contextualized discussion while maintaining scientific rigor and alignment with the available evidence.”

Round 2

Reviewer 1 Report (Previous Reviewer 3)

Comments and Suggestions for Authors

Authors have improved the manuscript. Figures quality must be improved.

Author Response

Comments 1: "Authors have improved the manuscript.”

Response 1: We sincerely thank the reviewer for the positive evaluation and for acknowledging the improvements made to the manuscript. We greatly appreciate the time and effort dedicated to reviewing our work and for the constructive feedback that has helped us enhance the quality of the study.

Comments 2: "Figures quality must be improved."

Response 2: We sincerely thank the reviewer for the comment. In this revised version of the manuscript, we have improved the quality of the figures as well as updated both tables to enhance their clarity and presentation.

Reviewer 2 Report (Previous Reviewer 1)

Comments and Suggestions for Authors

Dear authors,

Thanks for making constant and excellent efforts. I really appreciate the authors' effort.

Wish you all the best.

Author Response

Comments 1: "Thanks for making constant and excellent efforts. I really appreciate the authors' effort.”

Response 1: We sincerely thank the reviewer for their kind words and appreciation. We greatly value their constructive feedback throughout the process, which has been extremely helpful in improving the quality of the manuscript.

Comments 2: "Wish you all the best.”

Response 2: Thank you very much for your kind words and your valuable feedback throughout the entire review process. Your constructive comments and suggestions have been extremely helpful in enhancing the quality and clarity of our manuscript.
We sincerely appreciate the time and effort you have dedicated to the review and evaluation of our work.

This manuscript is a resubmission of an earlier submission. The following is a list of the peer review reports and author responses from that submission.

Round 1

Reviewer 1 Report

Comments and Suggestions for Authors

Thanks for inviting me to review the article titled “Simulated Clinical Stations in Quality and Patient Safety in a Primary Care Setting”. I reviewed this article with great interest. The manuscript has covered one of the important aspects related to patients’ safety in primary care. I appreciate the authors’ efforts. However, the following points are strongly in favor of my decision on the article.

  1. The objectives are unclear in the abstract and main text.
  2. Lack of theoretical framework in the introduction. Lacks the crucial things
  3. The authors did not follow the STROBE statement
  4. There are no clear research questions/hypotheses
  5. The authors need to explain a funnel-type introduction from broader aspects to narrow it to the rationale of the study and objectives/research questions/hypotheses.
  6. The manuscript lacks methodological rigor.
    1. The authors did not explain clearly about the study design. In a few sections, it seems descriptive in some sections, and in some sections, pre- and post-analysis.
    2. No proper explanation about the tool.
    3. No statistical test results are presented (e.g., p-values, effect sizes). Please include statistical analyses to confirm whether the observed improvements in knowledge scores are statistically significant. Also, the authors did not clarify the software tools, etc
  7. Results – Are there any particular reasons why key figures are included in the supplementary section? I consider it to be part of the main text.
  8. I consider the manuscript lacks generalizability. It can be shifted to the letter to the editor or the communication section.
  9. The discussion lacks in-depth analysis (even though the authors discussed some theories and existing studies).
  10. It is also crucial to discuss the limitations of the study.

Author Response

Comments 1: "The objectives are not clearly stated in the abstract and the main text."

Response 1: Thank you for pointing that out. I agree with this comment. Therefore, we have reinforced the objectives both in the abstract and in the introduction, explicitly highlighting the aim of assessing the impact of the training on knowledge and satisfaction.

Updated text:

“The main objective is to improve the competencies of Primary Care Team (PCT) professionals in managing critical and urgent situations, and to assess the impact of the intervention on knowledge and satisfaction using an immersive methodology based on active practice.” (abstract)

Comments 2: "The introduction lacks a theoretical framework. It is missing crucial elements."

Response 2: Agreed. We have therefore expanded the theoretical framework in the introduction, adding references on the use of simulation and its demonstrated effectiveness to emphasize this point.

Updated text:

“Multiple studies have shown that simulation-based training significantly improves clinicians’ skills and reduces errors […] that is, we refer to both technical and non-technical skills, including communication, leadership, and emotional pressure management, which are key elements in primary care settings.” (Introduction)

“Patient quality and safety are considered essential pillars in healthcare […]. Clinical simulation has been established as an effective methodology for improving these competencies, as indicated by several recent studies […].” (Introduction)

Comments 3: “The authors did not adhere to the STROBE statement”

Response 3: Thank you for this helpful comment.We have added a statement of adherence to STROBE in the Materials and Methods section.

Updated text:

“The study is based on a prospective pre-post observational design without a control group and follows the general guidelines of the STROBE statement for observational studies.” (Materials and Methods)

Comments 4: “There are no clear research questions or hypotheses.”

Response 4: Thank you for this valuable suggestion. We have incorporated the research hypotheses into the introduction.

Updated text:

“This study, led by the Primary and Community Care Directorate of the ICS, analyzes the impact of a training intervention using simulated clinical stations, with the aim of training Primary Care Team (PCT) professionals in the management of real critical situations, while promoting an active and meaningful learning environment that replicates routine clinical practice. The study simultaneously poses the hypothesis that clinical simulation effectively improves knowledge and increases professionals’ satisfaction, with the expectation of significant post-training improvements.” (Introduction)

Comments 5: “The authors should provide a funnel-shaped introduction, moving from broader aspects to the rationale of the study and the research objectives/questions/hypotheses.”

Response 5: We appreciate the reviewer’s thoughtful observation. We have restructured the introduction to move from general aspects toward the specific justification of the study.

Updated text:

“Patient quality and safety are considered essential pillars in healthcare […]. Clinical simulation has been established as an effective methodology for improving these competencies, as indicated by several recent studies […]. In this context, the need for improvement has led to the implementation of innovative strategies, such as clinical simulation, with the specific aim of assessing its impact in Primary Care Teams (PCTs).” (Introduction)

Comments 6: “The manuscript lacks methodological rigor.
• The authors did not clearly explain the study design. In some sections, it appears descriptive, while in others it suggests a pre-post analysis.
• There is no adequate explanation of the instrument used.
• Statistical test results (e.g., p-values, effect sizes) are not presented. Statistical analyses should be included to confirm whether the observed improvements in knowledge scores are statistically significant. Additionally, the authors did not clarify the software tools used, etc.”

Reviewer 2 Report

Comments and Suggestions for Authors

Thankyou for submitting your research article: Simulated Clinical Stations in Quality and Patient Safety in a Primary Care Setting. I have read the paper with interest. I am thinking there are many small glitches; areas where the study is not properly explained or it lacks documentation. I give details below. With improvements this paper can be made ready for publication

My first comment is: please revise the reference list to include the details of each citation’s publication (journal name, volume, page numbers, doi if available) and be consistent throughout. (Please see instructions in the author guide).

Although there is free format submission, your citations in text as a number following a word is unsettling; I suggest present citation numbers as a superscript?

I suggest these changes to improve the clarity of the paper:

Introduction line 56-58: suggest improve clarity in English: …the objective of this study is to conduct a program of simulation based education for an organization’s primary care team professionals and report learning outcomes. (the following text is part of the Methods;  “recreating different scenarios of daily clinical practice in order to evaluate the acquisition of knowledge and clinical practice.”

Methods: line 61 Heading 2.1 is preferred as ‘Sample population’

Line 67: please describe how many groups participated and the number of individuals in each group and their profession (or maybe summarize examples); It may be best to prepare a table summarizing groups and the participants in each?

Line 73: please translate to English

Line 87-95: I suggest a flow chart of simulation process, showing the list of individuals/observers and the simulation steps in the process? How many minutes in each simulation?

2.3. Evaluation and Analysis Process; This section should summarize the knowledge test (how many questions/open text or tick box, etc/who developed it/how long to complete…) and the satisfaction survey how many questions (give an example). Please also describe the analysis (missing).

Results line 133-34: what is the effect size of increases in knowledge? You should describe results in much more detail. I suggest the results charts are part of the paper rather than supplementary files??

Satisfaction is a key finding, I suggest quote the mean overall satisfaction scores for a couple of groups? And guide us to the chart. For any significant effects you should give the p=value.

Line 135: unclear whether you cite statistical means here?

Line 139-143 This is where effect sizes would be the best metric to describe differences between groups. Page 3-4 Lines 130-150 please revise the section of results comparisons with this in mind.

Discussion line 154: “has reduced the stress of professionals…” where is the evidence of this? Please state up front whether the study has addressed the stated objective and how?

Line 160: yes, debriefing needs to appear in your process table suggested earlier.

Line 168: Introduction of a mannikin? Please explain where introduced, etc. Please develop the discussion text related to your results section. I suggest you deal with each of the results in turn, in the order they are presented in the results section? The Discussion should be expressed more clearly. A limitations section should be added (strengths and limitations).

Conclusion: line 200-207 is not properly expressed: suggest: The importance of clinical training for all EAP professionals to manage clinical situations in practice must be emphasized. Training is required to enable practitioners to assume leadership roles in an acute clinical situation. The current program of simulation-based education has proven successful in providing EAP teams with greater knowledge and clinical skills practice.

Line 204: please summarize results- how many of the scenarios showed significant gains in knowledge?? The conclusion should be revised to flow better, with a report of the study outcomes and recommendations for the future?

Please revise the abstract in line with changes to the paper.

Please try to be more specific in your reporting.

Comments on the Quality of English Language

The English language is acceptable but some expected parts of description are missing.

Author Response

Comentaris 1: "Reviseu la llista de referències per incloure els detalls de la publicació de cada cita i assegureu-vos de la coherència en tot moment." 

Reviewer 3 Report

Comments and Suggestions for Authors

I have finished the review of the manuscript entitled "Simulated Clinical Stations in Quality and Patient Safety in a Primary Care Setting ", it related to a multidisciplinary educational intervention using simulation in Barcelona Spain.

The sample size is considerable and findings promising, nevertheless, Use of English need to be improved.

Rationale needs to be strengthened to specify the reasons, based in evidence and theoretical approximations, why you think that specific training would improve which specific competences, far from generic, it should be specific and pertinent.

Rather than providing training-based objectives, please state learning-based objectives, competence and/or attitude based.

Please mention which checklists were used and how they were validated, did you use an adapted version? please be specific and detail which capabilities and concepts were implicated.

Statistical analysis should be improved, given that there is no evidence provided that doctor, nurses, all other started equally, learn equally, etc., please provide by groups of professionals (e.g. doctors, nurses, other) initial knowledge points for each topic, checklist points per station by type of professional, average individual improvement using t tests and/or variance analysis across groups.

Ensure that the original objective is congruent to the measured dimensions, summarize and contrast learning in those dimensions mentioned in the objectives or classify by scenarios, and ensure that conclusions are congruent to the objective.

Mention the limitations of your study and areas of improvement, apart from cost-benefit.

Make sure that objective and conclusions in main text are congruent from those presented in the abstract.

Minor.

Are scenarios 3 & 4 the same? is scenario 3 a name? is it in Catalan?

Comments on the Quality of English Language

The structure of statements and paragraphs is presented in an unusual order.

Nouns like "action" seemed to be used as a substitute for "intervention" like between lines 81-86 or line 56

Some statements have redundant wording, like in line 65-66 "participants are ideally grouped into groups of doctors, nurses and administrative staff, with the addition of some other professionals from other professional categories who work in the EAPs" which is also written in present tense

Line 62 presents the trainees as "recipients" which in English would imply a passive attitude rather than an active learning.

There are many more. Professional Language is needed.

Also, 

If we considered the EQUATOR Gudelines, particularly, the Defined Criteria To Report INnovations in Education (DoCTRINE), Blanco M, Prunuske J, DiCorcia M, Learman LA, Mutcheson B, Huang GC (2022) in the Equator guidelines, the following aspects are to be considered:

Introduction

Y/N

1.      Need for the curriculum

N. Generic

2.      Review of relevant literature, theories, models, or published curricula

N. Partially

3.      Unique contribution of the curriculum to the literature

N. Not mentioned

Curriculum Development

Y/N

4.      Purpose/goals of the curriculum

Y. It is not learning-centered, but training-centered

5.      Outcome-based learning objectives

Y. Knowledge increases and acceptability?

6.      Target population of learners

Y. doctors, nurses and administrative staff, with the addition of some other professionals from other professional categories who work in the EAPs

Curriculum implementation

Y/N

7.      Instructional setting for curriculum delivery

Y. simulation center called CAP Balmes 63 that has been located in the basement of the corporate center in Barcelona

8.      Resources for implementing the curriculum

Y. simulated patients who are professional actors and actresses

simulation center called CAP Balmes. It should include mannequins with brand and model

9.      Description of instructional methods

simulated station of each of the 9 scenarios designed for which a standardized process is available.

Should include learning objective, educational rationale, why did you expect doctors, nurses and all other learn equally? Analysys should be devided my profession using anova or individual average increase using t-tests

10.  Methods to evaluate achievement of outcome-based learning objectives

N. Checklist (unspecified, unknown If it is validated)

2 knowledge tests pre-post tests

11.  Origin of evaluation instrument(s)

N. Cited only

Results

Y/N

12.  Number of learners participating in the curriculum

8916

13.  Number of participants included in the evaluation

8916

14.  Evidence of achievement of outcome-based learning objectives

Increase in knowledge and satisfaction, different to what was presented in objectives

Discussion

Y/N

15.  Summary of findings

N. Starts with a statement of quality and safety in primary care, not with summary findings

16.  Interpretation of findings in relation to the existing literature

N. Oriented mainly to multidisciplinary training using simulation, but no contrast to previous results in comparable interventions

17.  Lessons learned from the implementation of the curriculum

Y. Not specifically, just generic as a cost-benefit statement

18.  Limitations of the evaluation of the curriculum

N. None presented

19.  Describes future implications of the curriculum

N. Mentions how it should be continued, but not how to improve it

Author Response

Comments 1: "The justification needs to be strengthened by specifying the reasons—grounded in evidence and theoretical approaches—why you believe that specific training would improve particular competencies, which, rather than being generic, should be specific and relevant."
